# The Presence of Ascending Aortic Dilatation in Patients Undergoing Transcatheter Aortic Valve Replacement Is Negatively Correlated with the Presence of Diabetes Mellitus and Does Not Impair Post-Procedural Outcomes

**DOI:** 10.3390/diagnostics13030358

**Published:** 2023-01-18

**Authors:** Elke Boxhammer, Stefan Hecht, Reinhard Kaufmann, Jürgen Kammler, Jörg Kellermair, Christian Reiter, Kaveh Akbari, Hermann Blessberger, Clemens Steinwender, Michael Lichtenauer, Uta C. Hoppe, Klaus Hergan, Bernhard Scharinger

**Affiliations:** 1Department of Internal Medicine II, Division of Cardiology, Paracelsus Medical University of Salzburg, 5020 Salzburg, Austria; 2Department of Radiology, Paracelsus Medical University of Salzburg, 5020 Salzburg, Austria; 3Department of Cardiology, Kepler University Hospital, Medical Faculty of the Johannes Kepler University Linz, 4020 Linz, Austria; 4Department of Radiology, Johannes Kepler University Hospital Linz, 4020 Linz, Austria

**Keywords:** aortic valve stenosis, ascending aorta ectasia, biomarker, computed tomography, diabetes mellitus, TAVR

## Abstract

Both relevant aortic valve stenosis (AS) and aortic valve insufficiency significantly contribute to structural changes in the ascending aorta (AA) and thus to its dilatation. In patients with severe AS undergoing transcatheter aortic valve replacement (TAVR), survival data regarding aortic changes and laboratory biomarker analyses are scarce. Methods: A total of 179 patients with severe AS and an available computed tomography were included in this retrospective study. AA was measured, and dilatation was defined as a diameter ≥ 40 mm. Thirty-two patients had dilatation of the AA. A further 32 patients from the present population with a normal AA were matched to the aortic dilatation group with respect to gender, age, body mass index and body surface area, and the resulting study groups were compared with each other. In addition to echocardiographic and clinical characteristics, the expression of cardiovascular biomarkers such as brain natriuretic peptide (BNP), soluble suppression of tumorigenicity-2 (sST2), growth/differentiation of factor-15 (GDF-15), heart-type fatty-acid binding protein (H-FABP), insulin-like growth factor binding protein 2 (IGF-BP2) and soluble urokinase-type plasminogen activator receptor (suPAR) was analyzed. Kaplan–Meier curves for short- and long-term survival were obtained, and Pearson’s and Spearman’s correlations were calculated to identify the predictors between the diameter of the AA and clinical parameters. Results: A total of 19% of the total cohort had dilatation of the AA. The study group with an AA diameter ≥ 40 mm showed a significantly low comorbidity with respect to diabetes mellitus in contrast to the comparison cohort with an AA diameter < 40 mm (*p* = 0.010). This result continued in the correlation analyses performed, as the presence of diabetes mellitus correlated negatively not only with the diameter of the AA (r = −0.404; *p* = 0.001) but also with the presence of aortic dilatation (r = −0.320; *p* = 0.010). In addition, the presence of AA dilatation after TAVR was shown to have no differences in terms of patient survival at 1, 3 and 5 years. There were no relevant differences in the cardiovascular biomarkers studied between the patients with dilated and normal AAs. Conclusion: The presence of AA dilatation before successful TAVR was not associated with a survival disadvantage at the respective follow-up intervals of 1, 3 and 5 years. Diabetes mellitus in general seemed to have a protective effect against the development of AA dilatation or aneurysm in patients with severe AS.

## 1. Introduction

Dilatation of the AA—which has no real fixed numerical definition—can be detected cardiologically by transthoracic echocardiography or radiologically by thoracic computed tomography angiography (CTA). Instead, statements describing aortic ectasia with a 1.1-1.5-fold enlargement of the aortic diameter above the normal range can be found, whereas an aneurysm of the AA has been reported with an enlargement of > 1.5-fold [1]. However, the normal ranges depend on many key clinical factors, such as gender, age, height, body weight and consequently body mass index (BMI) or body surface area (BSA). Already in 2013, Rogers et al. [2] analyzed, as a subgroup of the worldwide-known Framingham Heart Study, the distribution and determinants of the thoracic aorta diameter in terms of its dependence on gender, age and BSA on the basis of a collective of 3431 patients and thus provided corresponding normal reference values.

In the COFRASA/GENERAC study [3], the extent of dilatation of the AA was investigated in 195 patients with AS, 29% of whom had an AA dilatation of ≥ 40 mm. Here, age, gender, BSA and valve anatomy (especially biscuspid aortic valves) were found to be relevant criteria for the occurrence of dilatation of the AA diameter. In contrast, the severity/gradient of AS did not correlate with potential dilatation of the AA.

Patients with both AS [4] and aortic valve insufficiency [5] may experience dilatation of the AA due to altered flow conditions. The level of risk for further clinical complications such as ruptures and dissections ultimately depends on the diameter, the condition of the wall of the AA, and further concomitant diseases.

Patients with severe AS, who suffer from multimorbidity and are therefore no longer candidates for surgical valve replacement, have had access to a minimally invasive procedure by means of TAVR in Europe for the past 15 years [6]. In patients with this constellation, the AA remains untouched regardless of the diameter and thus also in the case of dilatation.

Accordingly, the aim of the present study was composed of three parts:Can dilatation of the AA in TAVR patients be determined chemically in the laboratory by cardiovascular biomarkers, and can a pathophysiological link be established?Do patients with AS and AA dilatation who undergo TAVR have a survival disadvantage compared with patients with a normal AA?What clinical, echocardiographic or laboratory criteria/factors influence the diameter of the AA and potentially contribute to or even counteract ectasia?

## 2. Materials and Methods

### 2.1. Study Population

Figure 1 provides an overview of the study inclusion criteria using a flow chart.

The total study population included 221 patients with severe AS before TAVR in two large cardiology centers in Austria (Paracelsus Medical University Hospital Salzburg and Kepler University Linz), the data of which were collected from 2016 to 2018. Forty-two patients were excluded due to missing weight or height measurements, missing CT data or a lack of CT qualities, ultimately leaving 179 patients. Of these 179 patients, 32 patients had a diameter of the AA of ≥ 40 mm, which was most likely to meet the definition of aortic dilatation in the older collective presented here. Of the remaining patients with an AA diameter < 40 mm, 32 patients were manually matched for gender, body mass index and body surface area to the AA group ≥ 40 mm. The study protocol was authorized by the local ethics committees of Paracelsus Medical University Salzburg (415-E/1969/5-2016) and Johannes Kepler University Linz (E-41-16) and conducted in accordance with the principles of the Declaration of Helsinki and Good Clinical Practice. Written informed consent was available from all study participants.

### 2.2. Transthoracic Echocardiography

Common ultrasound devices such iE33 and Epiq 5 (Philips Healthcare, Hamburg, Germany) were used to perform transthoracic echocardiography in routine clinical practice before TAVR. Severe AS was categorized according to the current available guidelines of the European Society for Cardiology (ESC) using an AV Vmax (maximal velocity over aortic valve) of 4.0 m/s, an AV dpmean (mean pressure gradient over aortic valve) ≥ 40 mmHg and an aortic valve area ≤ 1.0 cm^2^ for definition. The left ventricular ejection fraction (LVEF) was computed by using Simpson’s method. Graduation of mitral, aortic and tricuspid valve regurgitation in terms of minimal, mild (I), moderate (II) and severe (III) was conducted by spectral and color Doppler images. Maximum tricuspid regurgitant jet velocity (TRV) was obtained by continuous wave Doppler imaging over the tricuspid valve.

### 2.3. CTA Protocol and Measurement of MPA Diameter for PH Assessment

The study patients at both centers—the University Hospital Salzburg and the Kepler University Linz—routinely received a pre-interventional electrocardiogram (ECG)-triggered CTA of the whole aorta until the proximal femoral arteries to assess, among others, the aortic annulus size, the aortic anatomy and vascular access. CTA scans were performed on multidetector CT scanners (Salzburg: Somatom Definition AS+, Siemens Healthcare, Erlangen, Germany; Linz: Brilliance 64, Philips Healthcare, Hamburg, Germany) with a patient-size-adapted tube voltage (80–120 kVp) and active tube current modulation. A bolus-tracking technique was applied with a 100 mL bolus of non-ionic iodinated contrast media followed by 70 mL saline solution injected at a flow rate of 3.5–5 mL/s.

For image analysis, a stationary workstation (Impax, Agfa-Gevaert, Mortsel, Belgium) was used. Two experienced investigators, blinded to all clinical and hemodynamic information, performed the measurement of the AA on corrected axial sections in mediastinal window settings. Finally, the mean of the obtained measurements of the AA diameter from investigators 1 and 2 was applied.

An illustration of the radiological measurement of the AA is shown in Figure 2.

Additionally, the quotient of AA/BSA was formed. Therefore, BSA was necessary using the DuBois formula (BSA = 0.007184 × Height^0.725^ × Weight^0.425^).

For the diagnosis of AA dilatation, a cut-off value of ≥ 40 mm was used. This criterion was not only considered as a good cut-off value in the common radiological and cardiological practice at the University Hospital Salzburg as well as the University Hospital Linz in the corresponding findings of thoracic CTA and transthoracic echocardiographies but could also be compared well with the relevant reference studies from Kerneis et al. [3] and Kim et al. [7], which were also based on this value.

### 2.4. Biomarker Analysis

Blood samples were obtained on the day of hospitalization and thus one day before the TAVR procedure using a vacuum-containing system. The collection tubes were centrifuged, and the plasma was separated from the blood components and then frozen at −80 °C. Samples were measured at equal time points under similar conditions.

Plasma levels of sST2, GDF-15, H-FABP, IGF-BP2 and suPAR were analyzed by using enzyme-linked immunosorbent assay (ELISA) kits (GDF-15: DY957, H-FABP: DY1678, IGF-BP2: DY674, suPAR: DY807, R&D Systems, Minneapolis, MN, USA). Manufactures’ instructions were performed for the adequate preparation of reagents. Therefore, serum samples and standard proteins were placed into the wells of ELISA plates (Nunc MaxiSorp flat-bottom 96-well plates, VWR International GmbH, Vienna, Austria) and incubated for two hours. The plates were treated with Tween 20/PBS solution (Sigma Aldrich, St. Louis, MO, USA). Afterwards, a biotin-labeled antibody was added and incubated for another two hours. A washing process was performed, and streptavidin–horseradish–peroxidase solution was added to the wells. A color reaction was generated after adding tetramethylbenzidine (TMB; Sigma Aldrich, USA). Optical density was determined at 450 nm on an ELISA plate reader (iMark Microplate Absorbance Reader, Bio-Rad Laboratories, Vienna, Austria).

### 2.5. TAVR Procedure

The indication for TAVR was conducted by a multidisciplinary team consisting of cardiologists and cardiac surgeons. The TAVR procedure was performed as previously described [8]. All patients received TAVR via transfemoral access.

### 2.6. Statistical Analysis

Statistical analysis with graphical representations was performed using SPSS (Version 25.0, SPSSS Inc., Armonk, NY, USA).

The Kolmogorov–Smirnov test was carried out to test the variables for normal distribution. Normally distributed metric data was expressed as mean ± standard deviation (SD), whereas non-normally distributed metric data was expressed as median and interquartile range (IQR). Comparison was conducted with Student’s unpaired *t*-test (normal distribution) and the Mann-Whitney U-test (non-normal distribution). Frequencies/percentages were used for categorial data and compared using the chi-square test.

Kaplan–Meier curves with the corresponding log–rank test were generated to determine whether there were differences in 1-, 3- and 5-year survival between patients with severe AS and an AA diameter ≥ 40 mm compared with an AA diameter < 40 mm.

Correlation analyses were performed using Pearson’s correlation coefficient when two metric variables were studied and Spearman’s rank–correlation coefficient when one of the two variables turned out to be ordinal or nominal to determine the strength between different AA measurements (AA ≥ 40 mm, AA, AA/BSA) and general patient characteristics. A *p*-value < 0.05 was considered statistically significant.

## 3. Results

### 3.1. Baseline Characteristics

Table 1 represents the most important clinical, echocardiographic, radiological and laboratory criteria of the included patients in relation to an AA < 40 mm and ≥ 40 mm.

A total of 32/179 patients had an AA diameter of ≥ 40 mm, which represented a percentage of 17.9%. By the manual matching performed, the included cohorts had almost the same age (81.97 ± 5.41 vs. 82.06 ± 5.01 years), a similar body weight (79.13 ± 12.78 vs. 80.34 ± 13.61 kg), a similar BMI (27.34 ± 3.84 vs. 27.12 ± 4.00 kg/m^2^) and a similar BSA (1.90 ± 0.17 vs. 1.93 ± 0.18 m^2^).

At 75% each, the proportion of men was clearly predominant in both groups. With regard to cardiovascular risk factors, the picture was balanced for arterial hypertension, coronary heart disease, atrial fibrillation, previous cardiac surgery, peripheral arterial occlusive disease and chronic obstructive pulmonary disease (COPD). However, a significant difference was found in the distribution of diabetes mellitus, revealing that patients with AA dilatation were significantly less likely to have diabetes mellitus (6.3%) than patients with a normal aortic diameter (31.3%; *p* = 0.010).

The echocardiographic criteria examined were also well balanced in both groups, with a tendency toward higher-grade aortic valve regurgitation in the ectasia group (*p* = 0.060).

### 3.2. Biomarker Concentrations

Figure 3 provides an overview of the corresponding plasma concentrations of the determined cardiovascular biomarkers depending on the diameter of the AA obtained (< 40 mm vs. ≥ 40 mm).

Typical cardiac biomarkers, such as cardiac troponin I, BNP, H-FABP and creatine kinase, which are routinely used in clinical practice, did not show relevant differences with respect to the diameter of the AA. In addition, not only relevant growth factors such as GDF-15 and IGF-BP2 but also inflammatory mediators such as suPAR and sST2 were not significantly increased or decreased in aortic ectasia.

Only hemoglobin was significantly (*p* = 0.046) increased, with values of 13.50 ± 2.08 vs. 12.70 ± 2.35 g/dL in the aortic dilatation group.

### 3.3. Kaplan–Meier Curves

Kaplan–Maier curves with regard to 1-year, 3-year and 5-year survival in terms of the dependence on the presence or absence of AA dilatation were performed and are presented in Figure 4.

At 1 year post TAVR (Figure 4A), 68.8% of patients with aortic dilatation and 62.5% of patients with a normal AA diameter were still alive. This corresponded to a log–rank test of 0.604 and thus no statistical significance was detected.

Similar ratios were seen at 3 years (Figure 4B) and 5 years (Figure 4C). After 3 and 5 years, 54.8% and 45.2%, respectively, were still alive in the dilated AA group, whereas 64.0% and 60.0%, respectively, of patients with normal aortic diameters survived. Again, the log–rank tests with *p* = 0.574 and *p* = 0.367, respectively, were without a relevant difference between the studied groups.

### 3.4. Correlation Analysis

Finally, to investigate the relationships between the measurements and quotients of AA (AA ≥ 40 mm, AA, AA/BSA) and the different clinical, echocardiographic and laboratory parameters, Pearson’s and Spearman’s correlation analyses was performed (Table 2).

Strikingly, an ascending aortic dilatation ≥ 40 mm correlated negatively with the presence of diabetes mellitus (r = −0.320; *p* = 0.010). This could be confirmed as well for the absolute AA diameter (r = −0.404; *p* = 0.001) and the indexing of the BSA (r = −0.350; *p* = 0.005).

The clearest positive correlation was between the numerical diameter of the AA and the left ventricular end-diastolic diameter (LVEDD), with r = 0.476 and *p* = 0.046, and a higher-grade aortic valve regurgitation, with r = 0.278 and *p* = 0.040.

## 4. Discussion

### 4.1. Biomarkers Examined Not Indicative of Vascular Remodeling

As can be seen from the present work, there was no satisfactory cardiovascular biomarker in the laboratory chemistry studied here that could support the radiological diagnosis of an enlargement of the AA and thus the pathophysiological vascular remodeling process.

GDF-15, as a member of the transforming growth factor (TGF)-β family, has been shown to have a relevant role in inflammatory processes and tissue injury and plays an important role in different cardiovascular and pulmonary pathomechanisms [9,10]. Regarding the effect on blood vessels and in particular on aortic structures, Sökmen et al. [11] revealed that GDF-15 correlates with the degree of aortic stiffness in patients with different degrees of hypertension and that increased “stiffening” of the aorta is associated with higher plasma levels of GDF-15. Both study groups examined here treated older TAVR patients with significantly increased hypertension, which was why progressive aortic stiffness could be assumed in both groups and could explain the lack of differences in the GDF-15 plasma levels.

A constellation similar to that of GDF-15 was probably also present in sST2. The group of Kim et al. [12] was able to show that the extent of aortic stiffness correlates positively with the laboratory detection of sST2 in patients with coronary angiography and invasively measured aortic pulse pressure for this biomarker, which is a member of the toll-like/integrin-like-1 receptor family and counteracts cardiac remodeling and fibrotic processes [13].

Regarding H-FABP as a clinically landmark cytoplasmic protein in the setting of acute myocardial injury [14,15] and IGF-BP2 as a relevant protein with cardioprotective function by downregulating the renin–angiotensin–aldosterone system [16,17], no comparative studies are available that ultimately support the data presented here. Finally, both H-FABP and IGF-BP2 were elevated in patients with severe AS, so that any vascular remodeling with dilatation of the AA was no longer important in the predominant pathophysiological process of increased pressure load and concentric myocardial hypertrophy.

suPAR, as an important biomarker for the detection of any inflammatory response in the human body [18], is handled differently in the current literature with regard to aortic processes. For example, Deng et al. [19] demonstrated in an animal experiment that suPAR was enhanced in angiotensin II-induced abdominal aortic aneurysms. This contrasts with a work by Böcskei et al. [20], who demonstrated an association between arterial stiffness and increased suPAR levels in COPD patients.

In summary, it is thus difficult to identify a cardiovascular biomarker that takes into account the pressure load due to AS with consecutive left ventricular hypertrophy and simultaneously detects the slowly progressive vascular dilatation that results from the relevant flow acceleration across the aortic valve.

### 4.2. Mortality after TAVR Is Not Affected by the Presence of AA Dilatation

Our results confirmed that dilatation of the AA is a common finding in patients undergoing TAVR and a common aortopathy in patients with AS in general, with incidence rates between 20% and 30% reported in the literature [3,21,22,23]. In our study, dilatation of the AA was present in nearly one-fifth of patients (32 of 179 patients; 17.8%), which was at the lower end of the incidence rates found in the publications mentioned above, but similar to the study of Ancona et al. [24]. A possible explanation is the different age profile of this dedicated TAVR cohort, as these patients were, on average, 22 years older when compared to patients undergoing conventional aortic valve replacements [25], and a share of the patients with AS might have been operated on an earlier age.

The data presented here additionally demonstrated that patients with severe AS and additional AA dilatation undergoing TAVR, in contrast to patients with a normal-sized AA, did not show any differences with regard to short-term survival (1 year) or long-term survival (3 and 5 years). By an appropriate matching process, these collected results were independent of gender, age, BMI and BSA.

These results were in agreement with a study of Rylski et al. [26], who had a very similar sample to that in the present work (71.0% vs. 75.0% male; 21.9% vs. 17.9% with ascending aortic dilatation ≥ 40 mm). It could be shown in a follow-up of 4 years that the survival of the patients did not depend on the extent of aortic dilatation and that the in-hospital mortality was not related to a dilated AA.

To date, there are only a few published results on diameter changes in dilated AA in patients who have undergone TAVR. The work of Lv et al. [27] demonstrated a slight reduction in AA diameter after TAVR in tricuspid aortic valve and mild aortic dilation (40–45 mm) subgroups, which was explained as a result of the correction of hemodynamic derangements caused by valve dysfunction. Non-progression or even a reduction in aortic dilatation could serve as a possible explanation of the non-significant differences in overall mortality and cardiovascular mortality in our cohort, which was also again in keeping with the study of Ancona et al. [24]. However, a (very slight) continuing AA dilation was demonstrated in a publication of He et al. [28] in a sample of comparable size to our study cohort.

### 4.3. Aortic Dilatation and Diabetes Mellitus Are Negatively Correlated

Our data showed a negative correlation of the aortic diameter (both indexed and not indexed to BSA) and the presence of diabetes mellitus, which suggested a protective effect of diabetes mellitus against AA ectasia (>40 mm) and aneurysm (>50 mm). In patients with diabetes mellitus, the mean aortic diameter was significantly smaller than in non-diabetic patients, and the prevalence of aortic dilatation was significantly higher in the non-diabetic subjects than in the patients with diabetes. These results are backed up by numerous works in the literature. In a very large Swedish nationwide observational study by Avdic et al. [29] that included almost 3 million individuals, patients with type 2 diabetes mellitus had significantly reduced risks of aortic aneurysm and aortic dissection as well as a reduced risk of mortality after hospitalization for aortic aneurysm. The presented data suggested a protective role of metabolic pathways that reduce the inflammatory response by the aortic wall and delay enlargement of the aorta [30]. Protection of the aortic wall through stronger glycated cross-links in vascular extracellular matrices in patients with glycemia and a hindered aortic root dilatation among patients with type 2 diabetes mellitus are only some of the suggested pathways responsible for this effect [31,32].

## 5. Conclusions

Aortic dilatation—defined as a diameter of the ascending aorta ≥ 40 mm—was not associated with a survival disadvantage in terms of short- or long-term survival in patients with severe AS undergoing TAVR. Diabetes mellitus in general seemed to have a protective effect against the development of ascending aortic ectasia or aneurysm in patients with severe AS.

## 6. Limitation

The present study was based on data from two cardiological Austrian centers over a circumscribed time period (2016–2018). The main limitation was the small sample size, which can be explained by the fact that patients with severe AS and concomitant aneurysm undergo surgery rather than TAVR and therefore did not appear in this study. Furthermore, technical pitfalls in echocardiographic and radiological measurements that lead to misclassifications should always be considered, even if examinations were performed by experienced clinical investigators.

## Figures and Tables

**Figure 1 diagnostics-13-00358-f001:**
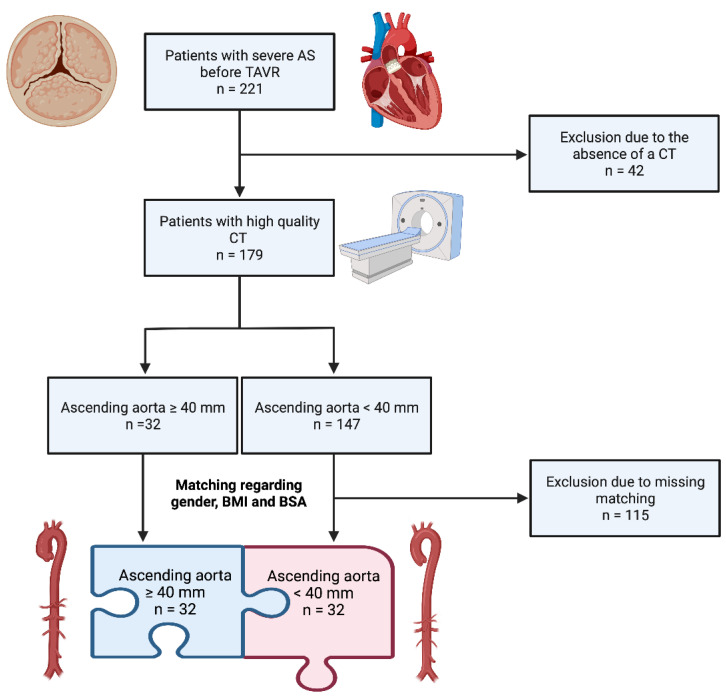
Flow chart with inclusion criteria of the study (created with BioRender.com). AS: aortic stenosis; CT: computed tomography; TAVR: transcatheter aortic valve replacement.

**Figure 2 diagnostics-13-00358-f002:**
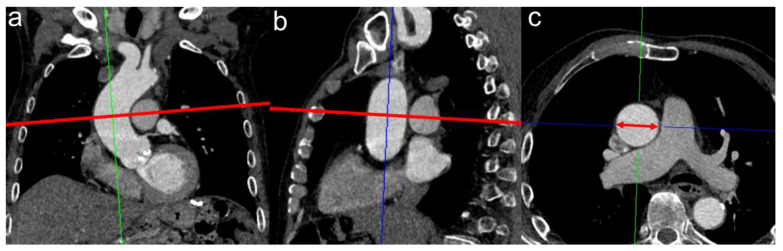
Measurement of the tubular ascending aorta at the level of the pulmonal artery bifurcation. Corrected axial vessel diameter was established using the multiplanar reconstruction plug-in of Deep Unity Diagnostics. The ascending aorta was manually corrected perpendicular to its centerline in coronal (**a**) and sagittal reconstruction (**b**). The obtained corrected axial aortic cross-section was measured (red double-headed arrow in (**c**)), and the widest diameter was used as maximum diameter.

**Figure 3 diagnostics-13-00358-f003:**
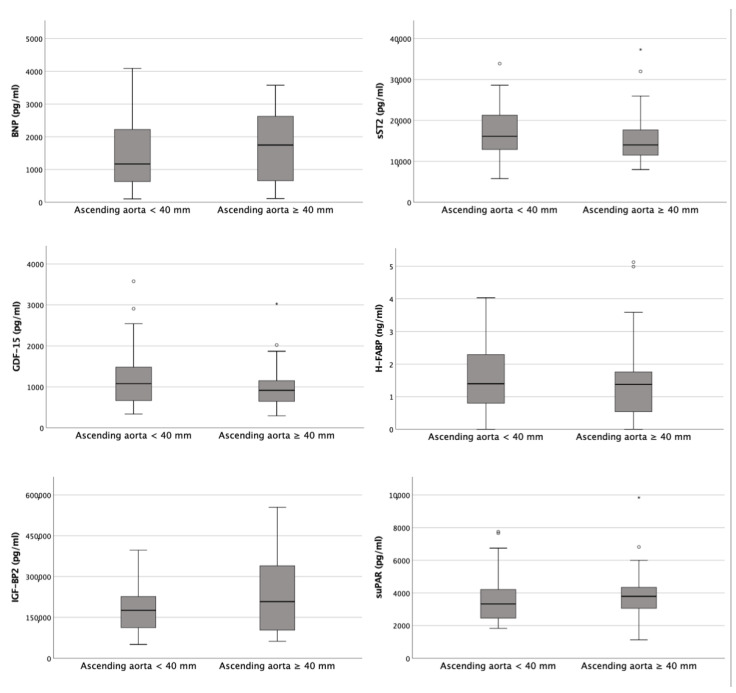
An overview of the corresponding plasma concentrations of the determined cardiovascular biomarkers depending on the diameter of the AA obtained (<40 mm vs. ≥40 mm).

**Figure 4 diagnostics-13-00358-f004:**
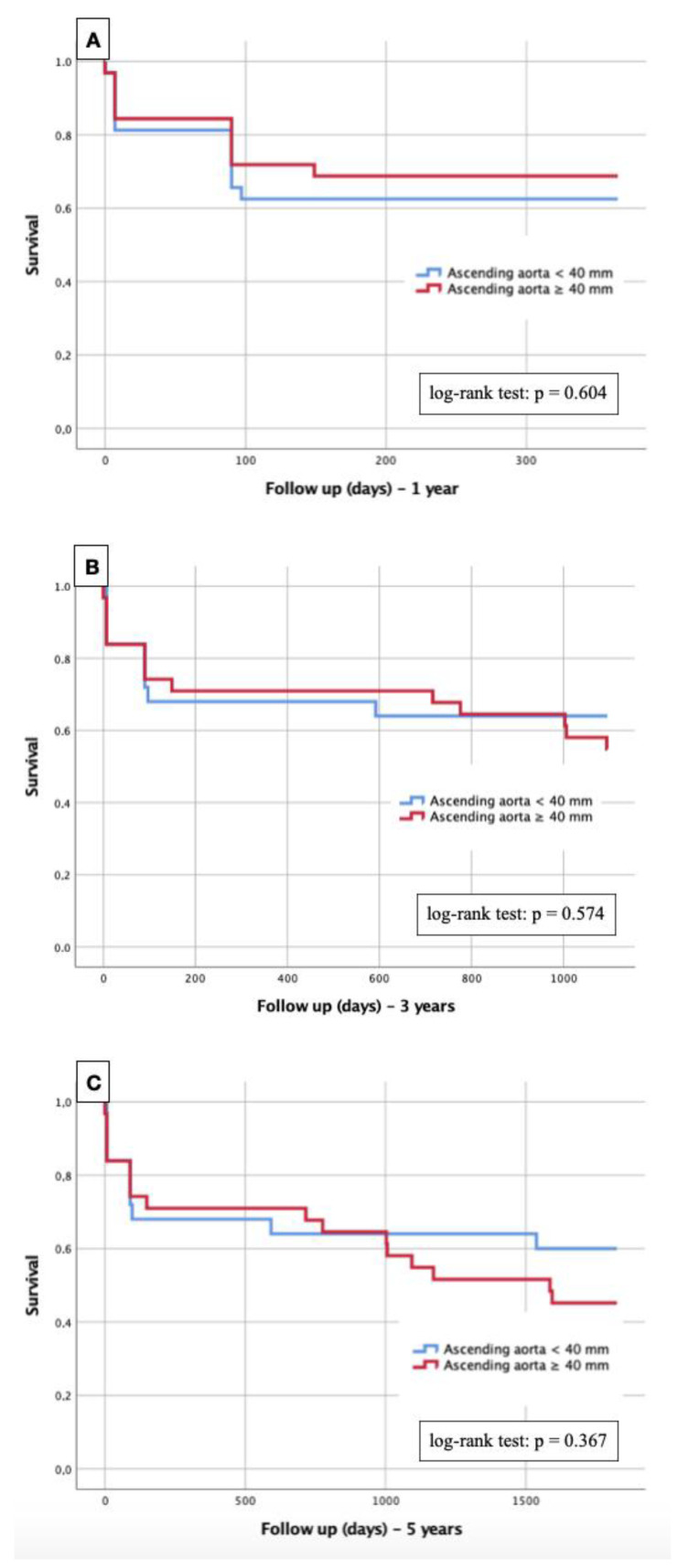
Kaplan–Meier curves for detection of 1-year- (**A**), 3-year- (**B**) and 5-year-survival (**C**) in terms of dependence on an AA diameter cut-off value ≥ 40 mm.

**Table 1 diagnostics-13-00358-t001:** The most important clinical, echocardiographic, radiological and laboratory criteria of the included patients in relation to an AA < 40 mm and ≥ 40 mm.

	Ascending Aorta < 40 mm n = 32	Ascending Aorta ≥ 40 mm n = 32	*p*
Clinical Data			
Age (years)—mean ± SD	81.97 ± 5.41	82.06 ± 5.01	0.943
Gender (male)—%	75.0	75.0	1.000
Weight (kg)—mean ± SD	79.13 ± 12.78	80.34 ± 13.61	0.713
Height (cm)—mean ± SD	169.94 ± 7.46	172.00 ± 8.31	0.300
BMI (kg/m^2^)—mean ± SD	27.34 ± 3.84	27.12 ± 4.00	0.823
BSA (m^2^)—mean ± SD	1.90 ± 0.17	1.93 ± 0.18	0.527
NYHA—median ± IQR	3.00 ± 0.75	3.00 ± 1.00	0.859
STSScore—mean ± SD	2.23 ± 1.61	2.26 ± 1.09	0.996
Concomitant Disease			
Diabetes mellitus—%	31.3	6.3	0.010
Arterial hypertension—%	78.1	65.6	0.266
CVD—%	81.3	71.9	0.376
CVD—1 vessel—%	25.0	21.9	0.881
CVD—2 vessels—%	9.4	3.1	0.334
CVD—3 vessels—%	9.4	6.3	0.697
Myocardial infarction—%	6.3	0.0	0.151
Atrial fibrillation—%	40.6	56.3	0.211
Previous cardiac surgery—%	9.4	9.4	1.000
Pacemaker—%	3.1	6.3	0.554
Stroke—%	6.3	3.1	0.329
PAOD—%	0.0	3.1	0.313
COPD—%	9.4	12.5	0.689
Smoking—%	11.2	14.6	0.588
CTA			
AA (mm)—mean ± SD	34.99 ± 2.48	42.24 ± 2.19	<0.001
AA/BSA (mm/m^2^)—mean ± SD	18.53 ± 1.86	22.13 ± 2.99	<0.001
Echocardiography			
LVEF (%)—mean ± SD	56.63 ± 12.15	52.41 ± 11.53	0.159
LVEDD (mm)—mean ± SD	44.25 ± 5.04	47.55 ± 5.77	0.221
IVSD (mm)—mean ± SD	16.00 ± 2.82	15.19 ± 2.79	0.271
AV Vmax (m/s)—mean ± SD	4.29 ± 0.60	4.34 ± 0.46	0.760
AV dPmean (mmHg)—mean ± SD	49.39 ± 13.93	46.44 ± 10.01	0.337
AV dPmax (mmHg)—mean ± SD	78.23 ± 20.07	75.66 ± 15.25	0.569
sPAP (mmHg)—mean ± SD	43.81 ± 14.16	43.39 ± 27.85	0.921
TAPSE (mm)—mean ± SD	19.54 ± 3.45	20.98 ± 3.71	0.317
AVI ≥ II—%	6.3	21.9	0.060
MVI ≥ II—%	9.4	15.6	0.421
TVI ≥ II—%	9.4	12.5	0.623
Laboratory Data			
Crea (mg/dL)—median ± IQR	1.10 ± 0.30	1.00 ± 0.28	0.326
BNP (pg/mL)—median ± IQR	1171.00 ± 1740.35	1749.50 ± 2010.08	0.492
cTnI (pg/mL)—median ± IQR	34.00 ± 73.75	27.00 ± 19.75	0.437
Hkt (%)—median ± IQR	38.20 ± 5.58	39.90 ± 2.08	0.103
Hb (g/dL)—median ± IQR	12.70 ± 2.35	13.50 ± 2.08	0.046
CK (U/L)—median ± IQR	73.00 ± 97.00	105.50 ± 73.75	0.479
sST2 (pg/mL)—median ± IQR	16,114.20 ± 8429.98	14,022.15 ± 6256.30	0.223
GDF-15 (pg/mL)—median ± IQR	1080.69 ± 871.92	916.977 ± 585.08	0.263
H-FABP (ng/mL)—median ± IQR	1.40 ± 1.62	1.38 ± 1.28	0.693
IGF-BP2 (pg/mL)—median ± IQR	176,048.92 ± 124,161.99	207,738.60 ± 276,393.81	0.750
suPAR (pg/mL)—median ± IQR	3322.47 ± 1935.84	3788.97 ± 1452.53	0.143

BMI: body mass index; BSA: body surface area; CTA: computed tomography angiography; CVD: cardiovascular disease; PAOD: peripheral arterial occlusive disease; COPD: chronic obstructive pulmonary disease; LVEF: left ventricular ejection fraction; LVEDD: left ventricular end-diastolic diameter; IVSD: inter-ventricular septal thickness at diastole; AV Vmax: maximal velocity over aortic valve; AV dpmean: mean pressure gradient over aortic valve; AV dpmax: maximal pressure gradient over aortic valve; sPAP: systolic pulmonary artery pressure; TAPSE: tricuspid annular plane systolic excursion; AVI: aortic valve insufficiency; MVI: mitral valve insufficiency; TVI: tricuspid valve insufficiency; Crea: creatinine; BNP: brain natriuretic peptide; cTnI: cardiac troponin I; Hkt: hematocrit; Hb: hemoglobin; CK: creatine kinase; sST2: soluble suppression of tumorigenicity-2; GDF-15: growth/differentiation of factor-15; H-FABP: heart-type fatty-acid binding protein.

**Table 2 diagnostics-13-00358-t002:** Tabular overview of correlation analysis (Pearson’s or Spearman’s) between AA ≥ 40 mm, AA diameter, AA/BSA ratio and various clinical characteristics.

Correlation	AA ≥ 40 mm		AA		AA/BSA	
	r	*p*	r	*p*	r	*p*
NYHA	−0.043	0.818	−0.046	0.804	−0.064	0.732
Diabetes mellitus	−0.320	0.010	−0.404	0.001	−0.350	0.005
Arterial hypertension	−0.139	0.273	−0.098	0.441	−0.143	0.260
CVD	−0.111	0.384	−0.193	0.126	−0.293	0.019
CVD—1	−0.019	0.884	0.006	0.966	−0.143	0.275
CVD—2	−0.125	0.342	−0.145	0.269	−0.019	0.884
CVD—3	−0.050	0.703	−0.157	0.231	−0.148	0.259
Myocardial infarction	−0.180	0.156	−0.193	0.127	0.061	0.633
Atrial fibrillation	0.156	0.217	0.255	0.042	0.132	0.298
Previous cardiac surgery	0.000	1.000	0.045	0.723	−0.036	0.776
Pacemaker	0.074	0.562	0.102	0.421	0.104	0.413
Stroke	0.124	0.337	0.129	0.317	0.168	0.191
PAOD	0.126	0.321	0.099	0.436	0.140	0.270
COPD	0.050	0.694	0.030	0.815	−0.009	0.941
LVEF	−0.179	0.157	−0.197	0.119	−0.208	0.100
LVEDD	0.227	0.364	0.476	0.046	0.202	0.422
IVSD	−0.144	0.277	−0.130	0.325	−0.244	0.063
AV Vmax	−0.005	0.973	0.114	0.413	−0.057	0.680
AV dpmean	−0.122	0.339	−0.102	0.426	−0.210	0.099
AV dpmax	−0.090	0.483	−0.032	0.806	−0.137	0.285
sPAP	−0.082	0.541	0.034	0.800	0.043	0.747
TAPSE	0.228	0.263	0.159	0.438	0.049	0.812
AVI ≥ II°	0.254	0.061	0.278	0.040	0.238	0.080
MVI ≥ II°	0.101	0.429	0.112	0.383	0.130	0.311
TVI ≥ II°	0.063	0.629	0.187	0.145	0.215	0.093
Crea	−0.124	0.329	−0.110	0.387	−0.148	0.242
BNP	0.092	0.497	0.055	0.685	−0.076	0.574
cTnI	−0.205	0.415	−0.286	0.249	−0.309	0.211
Hkt	0.206	0.103	0.135	0.287	−0.006	0.960
Hb	0.251	0.045	0.152	0.231	−0.023	0.858
CK	0.090	0.483	−0.082	0.524	−0.185	0.147
sST2	−0.156	0.226	−0.111	0.389	−0.148	0.252
GDF−15	−0.143	0.266	−0.129	0.318	−0.179	0.163
H−FABP	−0.051	0.696	−0.020	0.875	0.024	0.854
IGF−BP2	0.081	0.727	0.275	0.227	0.462	0.035
suPAR	0.187	0.145	0.002	0.989	0.167	0.195

AA: ascending aorta; BSA: body surface area; CVD: cardiovascular disease; PAOD: peripheral arterial occlusive disease; COPD: chronic obstructive pulmonary disease; LVEF: left ventricular ejection fraction; LVEDD: left ventricular end-diastolic diameter; IVSD: interventricular septal thickness at diastole; AV Vmax: maximal velocity over aortic valve; AV dpmean: mean pressure gradient over aortic valve; AV dpmax: maximal pressure gradient over aortic valve; sPAP: systolic pulmonary artery pressure; TAPSE: tricuspid annular plane systolic excursion; AVI: aortic valve insufficiency; MVI: mitral valve insufficiency; TVI: tricuspid valve insufficiency; Crea: creatinine; BNP: brain natriuretic peptide; cTnI: cardiac Troponin I; Hkt: hematocrit; Hb: hemoglobin; CK: creatine kinase; sST2: soluble suppression of tumorigenicity-2; GDF-15: growth/differentiation of factor-15; H-FABP: heart-type fatty-acid binding protein; IGF-BP2: insulin-like growth factor binding protein 2; suPAR: soluble urokinase-type plasminogen activator receptor; SD: standard deviation; IQR: interquartile range.

## Data Availability

The data presented in this study are available on request from the corresponding author.

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
