# Peer review of "The Presence of Ascending Aortic Dilatation in Patients Undergoing Transcatheter Aortic Valve Replacement Is Negatively Correlated with the Presence of Diabetes Mellitus and Does Not Impair Post-Procedural Outcomes"

_diagnostics, 2023, doi:10.3390/diagnostics13030358_

Round 1

Reviewer 1 Report

The study by Boxhammer et al. aimed to identify the effect of aortic valve stenosis (AS) and ascending aorta (AA) dilatation who underwent transcatheter aortic valve replacement (TAVR) compared with patients with normal AA. The results showed that the presence of AA dilatation before successful TAVR was not associated with a survival disadvantage at the respective follow-up intervals of 1, 3 and 5 years. However, I think that the content of this manuscript does not meet the standards of rigor required by the journal to be considered for publication.

Following are some concerns which need to be addressed by the authors.

1.     Overall sample size for the current study is low, thus the results might not be reliable given inadequate power of the study. An a priori power calculation would have been useful.

2.     Page 2, lines 76-84: The sentences should been removed.  

3.     No references list in the text. No discussion section.

Author Response

Dear reviewer 1,

you will find a detailed answer in the attached PDF file!

Thank you very much and best regards

Elke Boxhammer

Reviewer 2 Report

Comments and Suggestions for Authors

The manuscript is an original article that addresses the assessment of ascending aortic dilatation in patients undergoing transcatheter aortic valve replacement. The topic is interesting for researchers and clinicians, but it requires improvement by reviewing a few critical, major, and minor issues:

Critical

1.      The most glaring aspect is the lack of a discussion chapter. As such, if the authors will not introduce the discussions related to the imaging and laboratory findings in the context of the existing literature, the manuscript cannot be accepted.

Major

1.      The article is written in haste because paragraphs related to technical details of the editing were kept and were forgotten in the text (e.g. paragraphs between lines 76-84 and lines 260-263).

2.      In the parallel assessment performed by 2 investigators on the ascending aorta, it was not mentioned what the accepted difference is between the readings (% or absolute values), nor if the published values are the average of the readings.

3.      The authors did not consider smoking in patients' comorbidities, as a major risk factor. Please enter the data related to smoking from the 2 study groups in Table 1.

Minor

1.      Throughout the text there are numerous hyphens within the words whose places are not appropriate (e.g. line 3 transcath-eter, line 17 bi-omarker, line 18 retrospec-tive, etc…)

2.      In many places the authors did not put the references in square brackets.

3.      In Table 1, for the laboratory data, there is no adequate correspondence between the parameters and the numerical data, they being offset.

4.      Figure 3 is not necessary, because there were no statistically significant differences for those parameters; moreover it is redundant because the data are shown in Table 1.

After the amendment of the above comments in the manuscript, I would be in favor of publishing this paper.

Author Response

Dear reviewer 2,

you will find a detailed answer in the attached PDF file!

Thank you very much and best regards

Elke Boxhammer

Round 2

Reviewer 1 Report

Accept in present form

Reviewer 2 Report

I don't understand how such a different form of the manuscript could be submitted. The new manuscript sent is completely different. The authors have responded to my requirements and observations and made the related changes.

As such, I agree to the publication of the manuscript.
